# Suitability of a Low-Cost Wearable Sensor to Assess Turning in Healthy Adults

**DOI:** 10.3390/s22239322

**Published:** 2022-11-30

**Authors:** Rachel Mason, Joe Byerley, Andrea Baker, Dylan Powell, Liam T. Pearson, Gill Barry, Alan Godfrey, Martina Mancini, Samuel Stuart, Rosie Morris

**Affiliations:** 1Department Sport, Exercise and Rehabilitation, Northumbria University, Newcastle-upon-Tyne NE1 8ST, UK; 2Department Computer Science, Northumbria University, Newcastle-upon-Tyne NE1 8ST, UK; 3Department of Neurology, Oregon Health and Science University, Portland, OR 97239-3098, USA; 4Northumbria Healthcare NHS Foundation Trust, North Shields NE29 8NH, UK

**Keywords:** inertial sensors, turning, validation, wearables

## Abstract

*Background*: Turning is a complex measure of gait that accounts for over 50% of daily steps. Traditionally, turning has been measured in a research grade laboratory setting, however, there is demand for a low-cost and portable solution to measure turning using wearable technology. This study aimed to determine the suitability of a low-cost inertial sensor-based device (AX6, Axivity) to assess turning, by simultaneously capturing and comparing to a turn algorithm output from a previously validated reference inertial sensor-based device (Opal), in healthy young adults. *Methodology*: Thirty participants (aged 23.9 ± 4.89 years) completed the following turning protocol wearing the AX6 and reference device: a turn course, a two-minute walk (including 180° turns) and turning in place, alternating 360° turn right and left. Both devices were attached at the lumbar spine, one Opal via a belt, and the AX6 via double sided tape attached directly to the skin. Turning measures included number of turns, average turn duration, angle, velocity, and jerk. *Results*: Agreement between the outcomes from the AX6 and reference device was good to excellent for all turn characteristics (all ICCs > 0.850) during the turning 360° task. There was good agreement for all turn characteristics (all ICCs > 0.800) during the two-minute walk task, except for moderate agreement for turn angle (ICC 0.683). Agreement for turn outcomes was moderate to good during the turns course (ICCs range; 0.580 to 0.870). *Conclusions*: A low-cost wearable sensor, AX6, can be a suitable and fit-for-purpose device when used with validated algorithms for assessment of turning outcomes, particularly during continuous turning tasks. Future work needs to determine the suitability and validity of turning in aging and clinical cohorts within low-resource settings.

## 1. Introduction

Turning is a vital component of gait with over 50% of daily steps consisting of turns [1]. Turning requires the complex control of dynamic balance in a stable medial-lateral plane and coordination to re-orientate towards the new direction [2]. Turning characteristics including turn velocity, turn duration, number of steps in a turn, and smoothness during turning (jerk) which can be impaired in clinical populations including movement disorders and concussion [2,3,4,5,6,7]. Such research delineates the benefits of assessing turns in clinical populations, nonetheless previous mobility literature has mainly focused on straight-line gait analysis within laboratories [8,9], which may not fully represent functional impairments.

Assessment of comprehensive measures of turning are advantageous and provide detailed outcomes in both healthy and clinical populations [10,11]. However, conventional laboratory assessments within controlled environments can often lead to altered performance [12]. Traditional kinematic analysis methods such as optical motion capture systems allow for objective quantification of human movement [13] and are regarded as the ‘gold standard’ method [14,15]. However, such systems pose several barriers to implementation within clinical, and low-resource settings, including cost [16,17], access [18], and reduced ecological validity [5,19,20]. Accessible, affordable, and miniaturized wearable technology enables unobtrusive and continuous monitoring of turning in a range of environments (i.e., laboratory, home, community, clinic, etc.), providing a holistic overview of capabilities and relevant issues that can be explored [21,22]. Turning can be measured using a single (data logging) low-cost sensor on the waist, which can reduce patient burden and facilitates long term/continuous (7 day) monitoring of turning outside of clinical settings [21].

Previous work has demonstrated the validity of turning metrics obtained from a research-grade inertial sensor system/algorithm (Opal V1 and V2, APDM) compared to gold-standard motion capture [21,23]. Here, we sought to determine the suitability of a lower-cost option (in this case the AX6, Axivity) compared to the previous research-grade device, as more cost-effective alternatives could be used for pragmatic deployment in a range of settings. Therefore, the current study aimed to determine the suitability of the AX6 inertial sensor-based device against a validated approach during turning tasks in healthy young adults.

## 2. Methods

### 2.1. Participants

Thirty healthy young adults were invited to participate in this study. Participants were included in the study if; (1) they were aged between 18–40, and (2) they were able to stand and walk independently. Participants were excluded if they had comorbidities impacting on gait (e.g., muscular injuries). Prior to the study, participants were provided with an information sheet detailing the purpose and procedure of the study before providing written consent. Ethical approval was received from a Northumbria University Research Ethics Committee (Reference Number: 3672).

### 2.2. Demographic and Clinical Assessments

Age, height, and weight were recorded for all participants (Table 1). 

### 2.3. Equipment

Participants were fitted with the previously validated wearable reference tool (Opal inertial sensors 2000°/s gyroscope, magnetometer, and triaxial accelerometer sampling frequency of 128 Hz, version 1, APDM Wearable Technologies of Clario, Portland, OR, USA). In line with the previous turning algorithm validation studies [21,23], the reference device was attached at approximately the fifth lumbar vertebrae (L5) via a Velcro strap belt and synchronized with a laptop following each task. The AX6 (Axivity; 100 Hz accelerometer and 2000°/s gyroscope) sensor was taped at approximately L5 using double sided tape (Figure 1), under the Opal for direct comparison of algorithm performance. The time was recorded and synchronized with the laptop at the beginning and end of each task. After the three tasks were complete, data from the device were downloaded.

### 2.4. Gait Assessment 

Participants were asked to perform three tasks sequentially: Figure of eight turns course, two-minute walk, and 360° rotations clockwise and counter-clockwise. To familiarize the participant with the course, the assessor walked the participant through the turns course twice before allowing the participants to have a solo practice.
**Task 1:** The turns course included six turns per lap (Figure 2), the turns were comprised of two turns at 45°, 90° and 135° [24,25]. Each participant was asked to perform eight laps of the course at a pace comfortable to them (48 turns in total), participants were instructed to follow the tape markers of the course.**Task 2:** The second assessment involved participants walking at a comfortable speed back and forth between two lines set 5m apart. Participants were instructed to perform the 180° turn ‘as smoothly as possible’ at either end.**Task 3:** The final assessment consisted of the participant turning 360° clockwise and then counter-clockwise back and forth for two minutes in a fixed position. Participants were again asked to complete the turns as smoothly as possible.

### 2.5. Data Processing

At the commencement and conclusion of each task, the time was synchronized with the laptop and recorded by the assessor (i.e., time-stamp extraction). At the end of the trial, all device data were downloaded onto a laptop. Data (accelerometer and gyroscope) were processed for each device (Opal and AX6, Axivity Ltd, Newcastle upon Tyne, UK) separately using Matlab^®^ (2018R, Mathworks, MA, USA). The previously validated turning algorithm was used to process data from both sensors (Opal and AX6), with turns extracted from walking bouts within the three separate tasks [21,23]. Specifically, turns were detected using the horizontal (yaw) rotation rate of the waist sensor (Opal or AX6); a turn was detected when the yaw was >15°/s. A minimum of 35° trunk rotation around the vertical plane and a duration of 0.5–10 s was required for classification. The integration of the angular rate of the waist sensor around the vertical axis helped define the turn angles. Turning outcomes included the number of turns, average turn velocity (°/s), peak turn velocity (°/s), duration (s), turn angle (°), and jerk (°/s^2^). All turns were combined for the analysis.

### 2.6. Data Analysis

All data analysis was undertaken using SPSS^®^ (version 26, IBM, Armonk, NY, USA). Demographic characteristics were calculated as means and standard deviations (SD). Data were inspected through visual analysis of boxplots and followed by Kolmogorov–Smirnov test for normality. Intra-class correlation coefficients (ICC) were used to assess the absolute agreement between the Opal devices and the AX6. ICC values were classified based on research conducted by Koo and Li [26] and were as follows; Excellent (>0.90), good (0.75–0.89), moderate (0.50–0.74), and poor (<0.50). To demonstrate the bias within the limits of agreement (LoA), Bland–Altman plots were used [27]. Statistical significance was set at *p* < 0.05.

## 3. Results

### 3.1. Participant Demongraphics

A total of 30 participants completed the study (18 male and 12 female), on average participants aged 23.9 ± 4.9 years, see Table 1 for demographic details of participants. 

### 3.2. Turning Validation

Table 2 displays the descriptive data for turning characteristics from both the Opal and AX6 sensor, in addition to agreement between the two. Figure 3, Figure 4 and Figure 5 display an additional visual representation of agreement via Bland–Altman plots.

### 3.3. Task 1—Turning Course

Agreement between the outcomes from the AX6 and Opal was weakest during the turning course task with ICC values ranging between moderate and good agreement (Table 2 and Figure 3). Moderate agreement was shown for duration, turn angle and mean velocity (ICC 0.576 to 0.722, LoA% 18.2 to 30.7). Number of turns, peak velocity, and jerk displayed a good level of agreement for the static turning task (ICC 0.833 to 0.873, LoA% 19.0 to 26.5). 

### 3.4. Task 2—Two-Minute Walk

Agreement was stronger for the two-minute walk (2MW) task, with all gait characteristics, excluding turn angle displaying good agreement (ICC 0.824 to 0.888, LoA% 16.1 to 26.2 Turn angle ICC 0.683, LoA% 10.3). Agreement was strongest for the Jerk variable during the two-minute walk (ICC 0.888, LoA% 17.8) (Table 2 and Figure 4).

### 3.5. Task 3—Turning in Place

Agreement was strongest between the two sensors during the turning 360° in a fixed position task. In this task excellent agreement was shown for duration, turn angle, mean velocity, and jerk (ICC 0.906 to 0.989 & LoA% 5.2 to 14.2). Peak velocity displayed a good level of agreement for the static turning task (ICC 0.855 & LoA% 16.2) (Table 2 and Figure 5).

## 4. Discussion

This study investigated the suitability of a low-cost wearable inertial-based sensor (AX6) as a tool for assessment of turning characteristics in healthy young adults. Outcomes from the AX6 were compared to those from the previously validated research grade reference standard (Opal) [23]. Data from the AX6 and validated reference device showed good to excellent absolute agreement for turning analysis characteristics of turn number, duration, angle, peak velocity, mean velocity and jerk, comparable to previous validation studies using inertial sensors on the lumbar region to measure gait characteristics [28,29]. The general agreement of the devices was good to excellent, the turning in place for two-minutes task demonstrated the best agreement (ICC range = 0.850–0.990). The similar design, function and turn detection algorithm, stated previously in a study by El-Gohary et al. (2013), of the two devices explains these findings [23,30]. The discrepancies that were identified between the devices may be attributed to slight differences in sensor location. Both devices were placed by the L5 vertebrae, however, the AX6 was attached using double sided tape directly to the skin and the validated reference tool was attached using a belt, that may lead to unwanted movement. Device placement has been proven to have a significant effect on gait characteristics and variability [31], and therefore likely has similar impact on measures of turning, i.e., Jerk. The validated reference tool sensor displayed significantly higher mean jerk values in all 3 tasks, this could be attributed to looser fitting of the sensor straps causing additional vibrations of the device [32].

Characteristics of turning quality (duration, angle, velocity, and jerk) obtained poorer results. Turn duration displayed moderate agreement between sensors during the turns course task compared to good to excellent agreement displayed during the two-minute walk and turning in place tasks. The complexity of the tasks may account for this, the turns course utilizes three different turning angles (45°, 90°, and 135°) compared to the other two tasks which only incorporate one turning angle (Task 2—180°, Task 3—360°). Although valid identification of the start and stop times of turns does not necessarily correlate with accurate turn angle estimation [33], the reduced sampling rate of the AX6 (100 Hz) compared to the validated reference tool (128 Hz) may have led to less evident start and stops of turning, particularly for smaller 45° or 90° turns, thus influencing turn duration.

For turn angle, the AX6 tended to underestimate compared to the validated reference tool, with excellent agreement during the turning in place task. Previous research reinforces such findings, as when validating devices for turning that with increasing magnitude of turns came increasing accuracy [34]. An alternative explanation may be that the algorithm used in the current study detects turns at a minimum threshold of 35° and the smallest turn used in this study is 45°. It is possible that the participants performed these 45° turns at smaller turning angles leading to the sensors not accurately detecting these turns. Wearable sensors have been proven to mistakenly measure one larger turn as two smaller turns which may affect the turning quality measures [35]. Although, this is more likely to be seen in clinical populations such as Parkinson’s disease due to hesitations that may lead to slower turns [11]. This suggests further work needs to validate the low-cost sensor in clinical populations.

For the number of turns detected, the magnitude of the turn does not seem to have had an effect. The agreement was excellent in task 3 which included the largest turns (360°) followed by task 1 showing good agreement for a range of smaller turns (45°, 90°, and 135°) and moderate in task 2 which involved medium sized turns (180°). The original validation of the algorithm using the validated reference tool demonstrated a sensitivity of 0.90 and 0.76 when compared with motion capture analysis and video rater and a specificity of 0.75 and 0.65 [23]. This lower specificity value suggests the tendency of the algorithm to falsely record a turn. This is consistent with the present finding that the AX6 sensor predicted larger mean number of turns in each task (e.g., 23.0 vs. 21.9). Detection of additional spurious turn in the AX6 data compared to the Opal sensor may be due to errors relying on observer timestamp recording to start and end the trial with the AX6 (i.e., manual data extraction), which is an automated process by the reference tool sensor start and stop software. Previous studies have shown that manual recording of mobility is prone to errors in starting and stopping recording when compared to inertial sensors [36,37].

### 4.1. Limitations

The primary aim of this study was to determine the suitability of a cost-effective inertial-based device for turning tasks. However, there are several limitations that need to be considered. First, our study only analyzed the agreement within a small sample of healthy young adults; therefore, findings cannot be assumed to be applicable to clinical or older adult populations. Second, supplementary research is necessary to determine whether the AX6 devices (or other similar lower cost inertial measurement units) are scalable to more natural environments involving wider populations. Third, the sensors in this study were not attached together and therefore we may have seen some interference within our analysis. Finally, the wearable sensor requires the data collected to be downloaded and manually analyzed, therefore may currently be inapplicable for a clinical setting due to the need for trained personnel. Albeit it is a step towards implementing wearable technology for turning analysis in a such settings.

### 4.2. Implications 

The potential for wearable sensors to be adapted for use in a variety of populations and settings has previously been explored [17]. Numerous studies have indicated that wearable devices can be utilized to highlight gait characteristics that distinguish clinical populations such as mTBI [38] and PD [39] from control subjects, as well as assess fall risk [22,40,41]. Following further validation in clinical populations as well as in free-living environments, the AX6 may provide a cost-friendly, objective method of gait analysis in low-resource settings.

## 5. Conclusions

The AX6 demonstrated good to excellent agreement for turning characteristics including number of turns, peak velocity, mean velocity, and jerk. The AX6 provides a valid measure of turning in a laboratory setting for healthy young participants. Validation of turning characteristics using the AX6 Axivity sensor is now required in aging and clinical populations, as well as the home and community environment to provide habitual outcome measures. The affordability of the AX6 provides an opportunity for low-resource settings to uptake wearable sensors for use in larger populations.

## Figures and Tables

**Figure 1 sensors-22-09322-f001:**
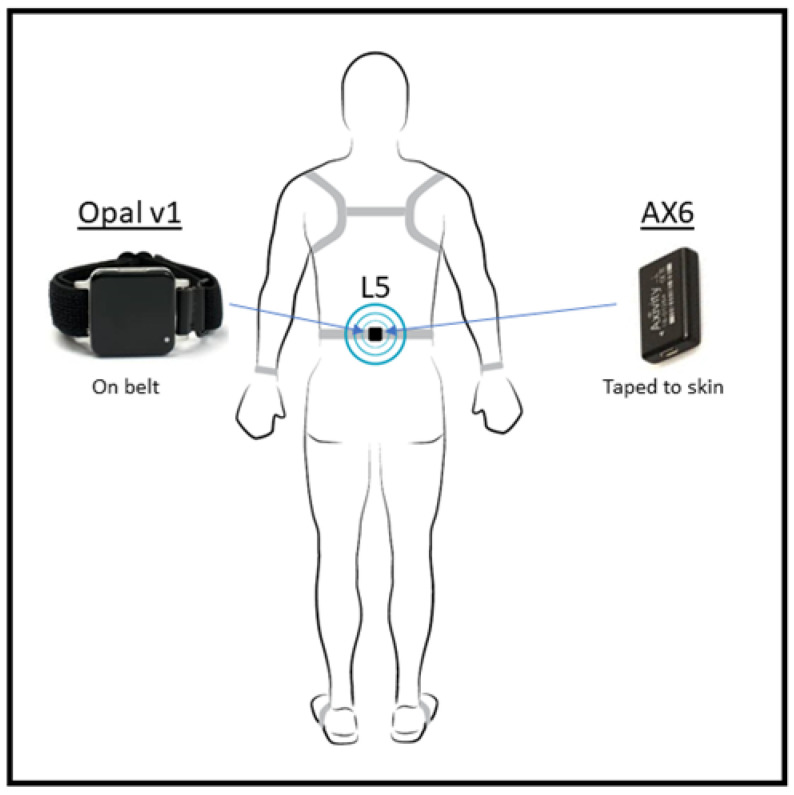
Participant set-up.

**Figure 2 sensors-22-09322-f002:**
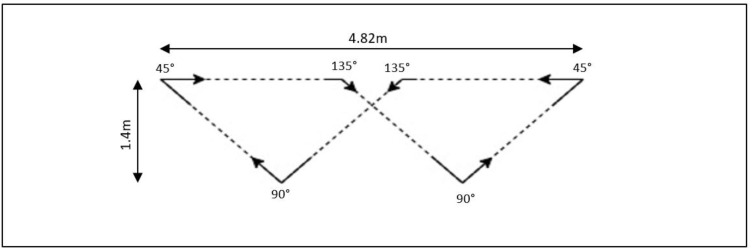
The turning course for Task 1.

**Figure 3 sensors-22-09322-f003:**
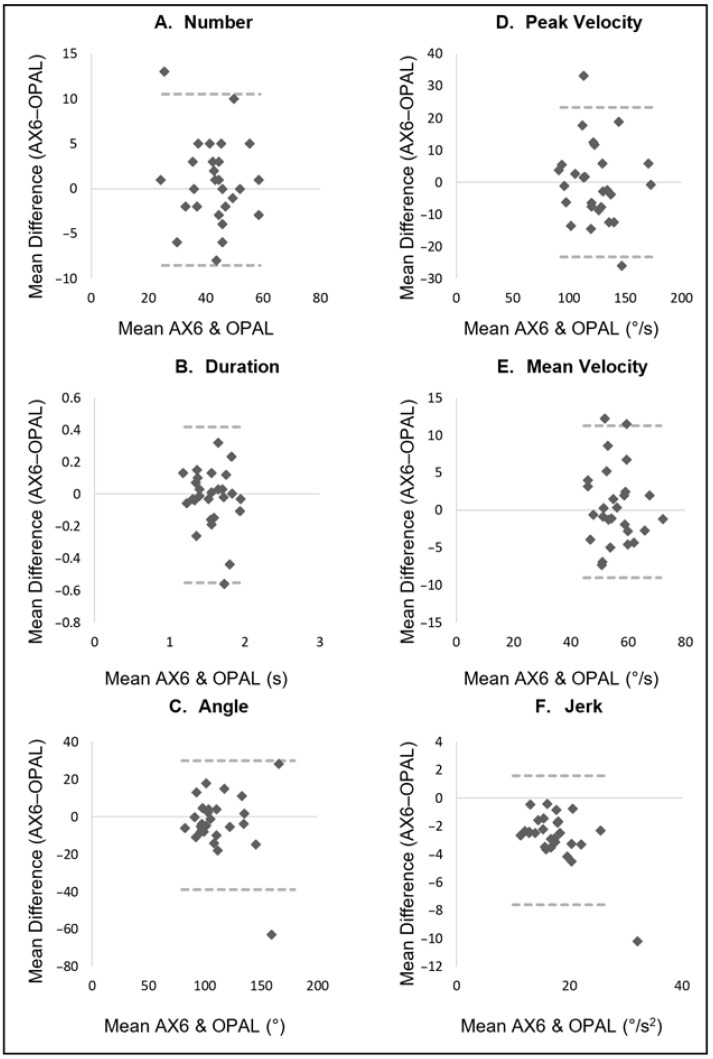
Turning Course task: Bland–Altman plots displaying agreement between AX6 and OPAL for (**A**) Number, (**B**) Duration, (**C**) Angle, (**D**) Peak velocity, (**E**) Mean velocity, (**F**) Jerk.

**Figure 4 sensors-22-09322-f004:**
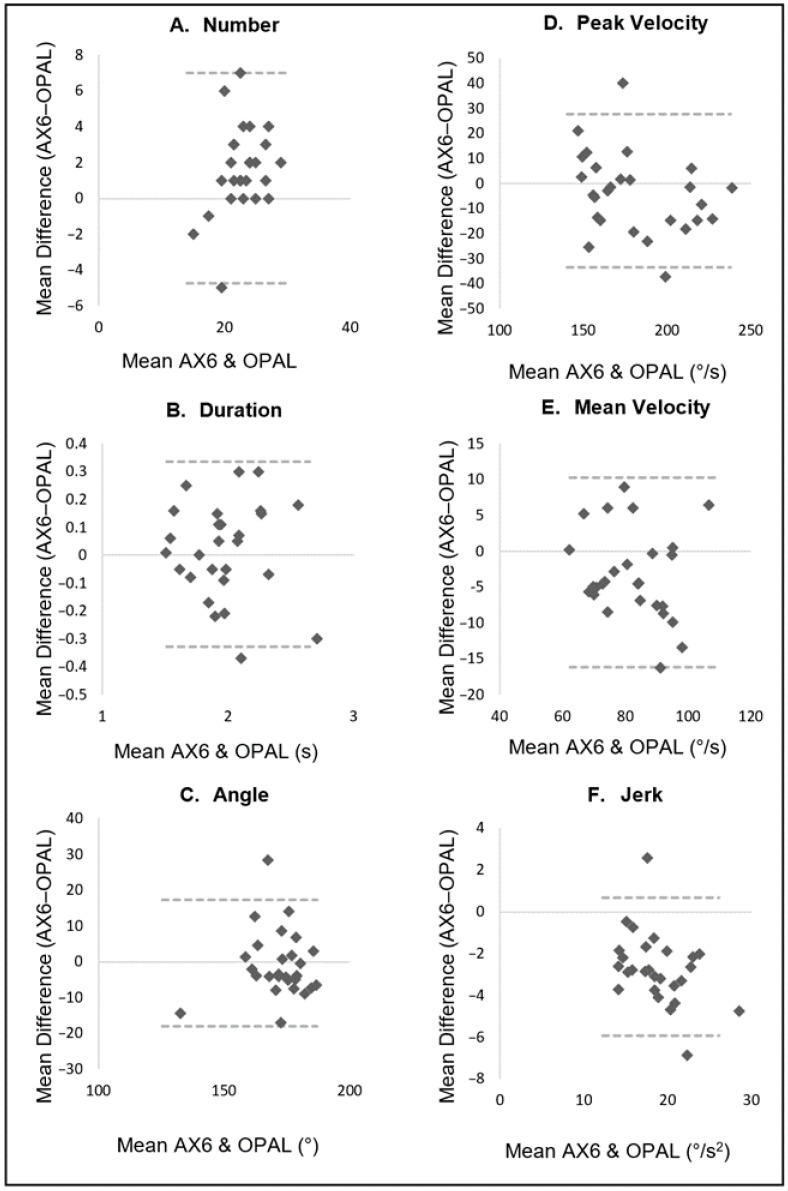
Two-Minute Walk task: Bland–Altman plots displaying agreement between AX6 and OPAL for (**A**) Number, (**B**) Duration, (**C**) Angle, (**D**) Peak velocity, (**E**) Mean velocity, (**F**) Jerk.

**Figure 5 sensors-22-09322-f005:**
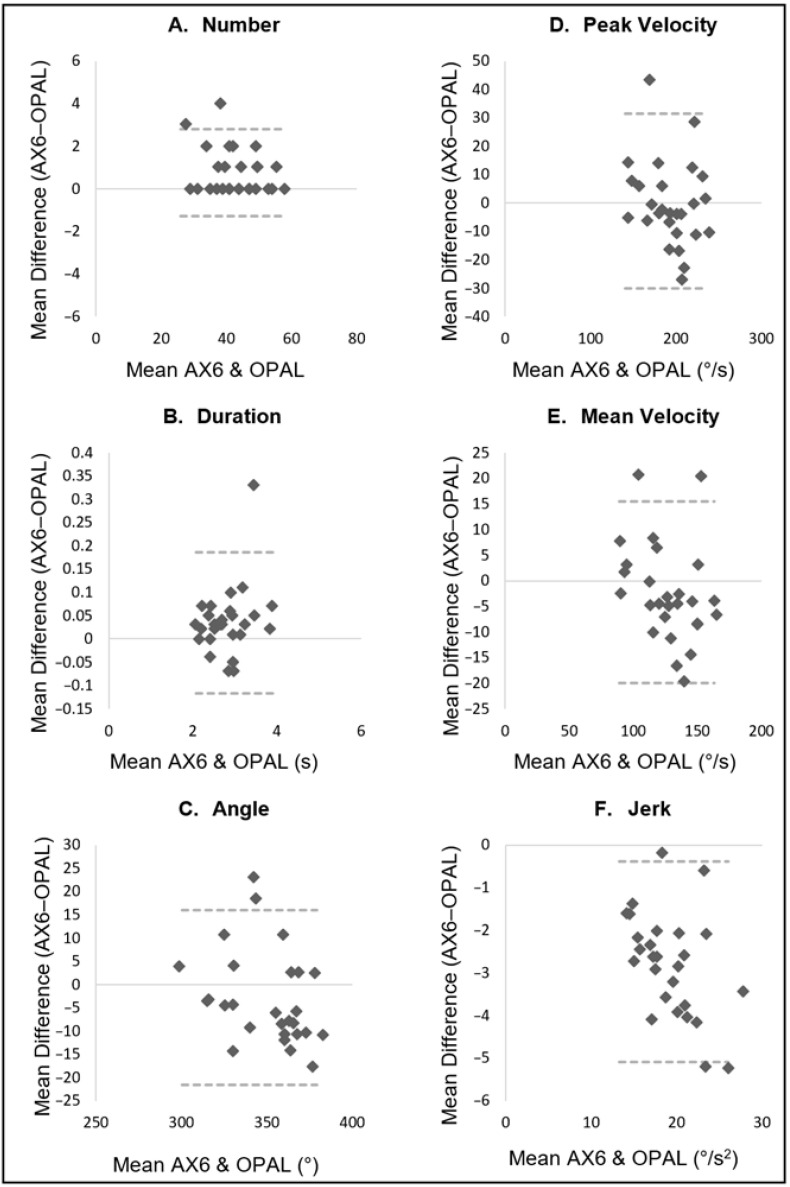
Turning in Place task: Bland–Altman plots displaying agreement between AX6 and OPAL for (**A**) Number, (**B**) Duration, (**C**) Angle, (**D**) Peak velocity, (**E**) Mean velocity, (**F**) Jerk.

**Table 1 sensors-22-09322-t001:** Demographic characteristics of participants.

	Male (n = 18)	Female (n = 12)
Age (years)	23.6 ± 5.4	23.7 ± 4.0
Height (cm)	175.2 ± 9.5	174.8 ± 6.3
Weight (kg)	77.3 ± 12.5	76 ± 12.1

**Table 2 sensors-22-09322-t002:** Mean difference, ICC_2.1_, mean difference (*p*), limits of agreement (LOA%), and Pearson correlation between OPAL and AX6 sensors for turning characteristics for three turns tasks.

Task	Turn Characteristics	AX6 (n = 30)	OPAL (n = 30)	Agreement								
		Mean (SD)	Mean (SD)	Mean Difference	ICC	Lower Bound	Upper Bound	*p*	LoA (%)	LoA95%	Pearson *r*	Pearson *p*
**Turning Course**	Number	43.33 (8.98)	42.37 (10.27)	−0.97	0.873	0.751	0.937	<0.285	22.2	9.526	0.881	<0.001
	Duration (s)	1.55 (0.21)	1.61 (0.31)	0.07	0.576	0.277	0.773	<0.156	30.6	0.484	0.618	<0.001
	Angle (°)	110.03 (20.82)	114.47 (26.00)	4.44	0.722	0.493	0.857	<0.177	30.7	34.427	0.740	<0.001
	Peak Velocity (°/s)	123.01 (19.78)	123.00 (21.43)	−0.01	0.833	0.679	0.917	<0.997	19.0	23.330	0.836	<0.001
	Mean Velocity (°/s)	56.38 (6.85)	55.23 (6.89)	−1.15	0.716	0.484	0.854	<0.231	18.2	10.139	0.716	<0.001
	Jerk	15.79 (3.86)	18.80 (5.25)	3.00	0.872	0.748	0.937	<0.01	26.5	4.575	0.914	<0.001
**2MW**	Number	23.03 (3.95)	21.90 (2.98)	−1.13	0.632	0.356	0.806	<0.048	26.2	5.885	0.657	<0.001
	Duration (s)	1.98 (0.30)	1.98 (0.30)	0.00	0.840	0.691	0.921	<0.906	16.8	0.332	0.840	<0.001
	Angle (°)	172.10 (11.49)	172.47 (11.20)	0.37	0.683	0.432	0.835	<0.823	10.3	17.725	0.683	<0.001
	Peak Velocity (°/s)	178.79 (25.33)	181.63 (30.03)	2.84	0.842	0.694	0.921	<0.329	17.0	30.638	0.854	<0.001
	Mean Velocity (°/s)	80.71 (10.69)	83.67 (11.97)	2.96	0.824	0.662	0.912	<0.022	16.1	13.210	0.829	<0.001
	Jerk	17.21 (3.23)	19.84 (3.84)	2.63	0.888	0.778	0.945	<0.01	17.8	3.292	0.901	<0.001
**Turning in place**	Number	42.10 (8.31)	41.33 (8.51)	−0.77	0.992	0.984	0.996	<0.01	4.9	2.038	0.993	<0.001
	Duration	2.91 (0.54)	2.88 (0.52)	−0.03	0.989	0.978	0.995	<0.02	5.2	0.152	0.990	<0.001
	Angle	349.94 (20.60)	352.72 (23.51)	2.78	0.906	0.811	0.954	<0.124	5.4	18.833	0.913	<0.001
	Peak Velocity (°/s)	190.62 (27.41)	189.84 (30.85)	−0.77	0.855	0.718	0.928	<0.790	16.2	30.779	0.861	<0.001
	Mean Velocity (°/s)	123.57 (21.43)	125.77 (24.29)	2.20	0.922	0.843	0.962	<0.193	14.2	17.750	0.929	<0.001
	Jerk	17.84 (3.33)	20.57 (3.80)	2.73	0.944	0.885	0.973	<0.01	12.2	2.352	0.952	<0.001

## Data Availability

Data is available on request.

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
