# Peer review of "Suitability of a Low-Cost Wearable Sensor to Assess Turning in Healthy Adults"

_sensors, 2022, doi:10.3390/s22239322_

Round 1

Reviewer 1 Report

A schematic of the location of the sensor AXL is needed.

The variables need to be explained since the variance is a number it is hard to know what is "the usual" reading - can it be done in percentage instead?

Very creative to try to find a better alternative.

Author Response

We thank the Reviewers for their time and efforts in reviewing our manuscript. We have made changes in the manuscript highlighted in yellow and provide detailed responses to comments in the tables below.

Reviewer 1

Reviewer comment

Response

A schematic of the location of the sensor AXL is needed.

This has been added into the methods section (Figure 1).

The variables need to be explained since the variance is a number it is hard to know what is "the usual" reading - can it be done in percentage instead?

The turning variables that were collected and reported are explained in section 2.5, as well as in the previous algorithm validation studies that we reference [21,23].

Very creative to try to find a better alternative.

We thank the reviewer for their kind comment.

Reviewer 2 Report

Unfortunately that the comparison target was limited to only one result. This work draws the conclusions without comparing it to more various works or setting different experimental conditions.

1. The authors emphasized that they chose the AX6 as a lower cost option. The current price of the AX6 is £189. However, there are many cheaper accelerometer, IMU sensors, and gravity sensors out there. It seems that other products with similar or lower price should also be included for the comparison. I understood that authors were trying to compare the 6-axis sensor (AX-6) with the 9-axis sensor (Opal). It would be helpful if the results of experiments using other 6-axis sensors instead of the AX-6 are presented

2. It will be easier to understand if the authors show the AX6 sensor attached to the body with a picture or photo. Also, please explain how the authors determined that the position was the best position to detect turning. It is necessary to include a photo so that one can easily see where the AX6 is attached the subject. In gait analysis, where the sensor is attached to the subject affects the measurement result. So, I think it is important to show the photo (or figure) to interested readers.

3. Also, please explain how authors determined that the position was the best position to detect turning.

4.Why didn't the authors test with more than one AX6 attached? There is no mention of whether the Opal is worn on the left foot and the AX-6 is worn on the right foot. I don't know if the subjects wear both devices on the same foot. Please clarify it, And, It is necessary to present the experimental results of wearing the AX-6 on the left and right feet, respectively.

Author Response

We thank the Reviewers for their time and efforts in reviewing our manuscript. We have made changes in the manuscript highlighted in yellow and provide detailed responses to comments in the tables below.

Reviewer 2

Reviewer comment

Response

Unfortunately that the comparison target was limited to only one result. This work draws the conclusions without comparing it to more various works or setting different experimental conditions.

This work is a validation study of a previously developed algorithm being used within a low-cost sensor and compared to the previous high-cost sensor.

Multiple conditions are examined for the turning outcomes, such as walking back and forth, turning in place and a turning course, which are performed to represent ‘real-world’ turning. This follows on from the previous validation study that performed several of these tasks (El-Gohary et al. 2016).

There are also multiple outcomes compared to the turning performance; number, peak velocity, mean velocity, duration, angle, jerk. Therefore, validation is not based upon a single result or outcome.

1. The authors emphasized that they chose the AX6 as a lower cost option. The current price of the AX6 is £189. However, there are many cheaper accelerometer, IMU sensors, and gravity sensors out there. It seems that other products with similar or lower price should also be included for the comparison. I understood that authors were trying to compare the 6-axis sensor (AX-6) with the 9-axis sensor (Opal). It would be helpful if the results of experiments using other 6-axis sensors instead of the AX-6 are presented

The AX6 wearable sensor (~£186) is substantially lower cost that the Opal device (~£2000 per sensor). The cost is a huge factor in the adoption of technologies within clinical trials and healthcare settings.  While there are wearables that may be lower cost, the Axivity sensors (AX3 and AX6) have been extensively used for mobility assessments within various clinical groups, being CE marked and applicable for clinical trials / research.

While the Opal may include an accelerometer, gyroscope, and magnetometer, and the AX6 may only have an accelerometer and gyroscope, the algorithm that was being applied only used the gyroscope from the sensors, which was recorded using the same settings. Therefore, adding a separate 6-axis sensor would not provide any further details, and is not within the scope of this manuscript. We aimed to validate the previously developed algorithm within the AX6 sensor, to ensure we can accurately measure turning from this sensor.

2. It will be easier to understand if the authors show the AX6 sensor attached to the body with a picture or photo. Also, please explain how the authors determined that the position was the best position to detect turning. It is necessary to include a photo so that one can easily see where the AX6 is attached the subject. In gait analysis, where the sensor is attached to the subject affects the measurement result. So, I think it is important to show the photo (or figure) to interested readers.

A figure showing the position/location of the AX6 has been added into the methods section (Figure 1).

3. Also, please explain how authors determined that the position was the best position to detect turning.

The lumbar region was used as this was the location where the previously developed algorithm was validated. Typically for gait or mobility assessments the lumbar region is used due to the relation to center of mass, and ability to detect steps / asymmetry of steps from this location. We did not change locations as this would invalidate the algorithm.

This has been added into Section 2.3 and now reads: ‘In line with the previous turning algorithm validation studies [21, 23], the reference device was attached at approximately the fifth lumbar vertebrae (L5) via a Velcro strap belt and synchronized with a laptop following each task. The AX6 (Axivity; 100Hz accelerometer and 2000°/sec gyroscope) sensor was taped at approximately L5 using double sided tape (Figure 1), under the Opal for direct comparison of algorithm performance.’

4.Why didn't the authors test with more than one AX6 attached? There is no mention of whether the Opal is worn on the left foot and the AX-6 is worn on the right foot. I don't know if the subjects wear both devices on the same foot. Please clarify it, And, It is necessary to present the experimental results of wearing the AX-6 on the left and right feet, respectively.

Both devices were attached at the lumbar spine (L5), one Opal via a belt, and the AX6 via taped directly to the skin. The devices were not worn on the feet.

The lumbar region was used as this was the location where the previously developed algorithm was validated. Typically for gait assessments the lumbar region is used due to the location in relation to center of mass, and ability to detect steps / asymmetry of steps from this location.

Reviewer 3 Report

The paper is well-presented and the experiments well-conducted and reported. One small issue that the references need to be arranged in order of citation.

Detailed comments:

How is the current study different from the article https://www.sciencedirect.com/science/article/pii/S2352648321000428 ,  the authors need to clearly state the motivation of the work and the differences from existing published works.

The bland-Altman plots need to be made clearer to reflect agreement or not along with the quantitative measures.

Table 2, the Pearson correlations, make headings consistent.

The authors need proper justification for the comparison with AX6 device. Simply stating the cost is not reflective of the quality of the measurement produced by the device. Trying more devices would add more reliability to the evaluation.

Author Response

We thank the Reviewers for their time and efforts in reviewing our manuscript. We have made changes in the manuscript highlighted in yellow and provide detailed responses to comments in the tables below.

Reviewer 3

Reviewer comment

Response

The paper is well-presented and the experiments well-conducted and reported. One small issue that the references need to be arranged in order of citation.

Thank you for your comments. This has been addressed and the paper is now referenced in the style

‘MDPI ACS Journals

Citation Style: Non-superscripted Number’

How is the current study different from the article https://www.sciencedirect.com/science/article/pii/S2352648321000428 ,  the authors need to clearly state the motivation of the work and the differences from existing published works.

This work is different from our previous article (that the reviewer linked) on the premise that existing published work focuses on gait metrics (e.g., step/stride time, swing time and stance time), primarily during straight line tasks (e.g., 4m walk test), derived from the accelerometer within the AX6, which was compared to the AX3 sensor. Although turn metrics were calculated during the timed up and go with the AX6 sensor, there was no reference method to validate turning metrics (of which there were only two metrics).

The work presented here adds to the current body of knowledge by investigating turn metrics (several more metrics than previous work) through testing a previously developed and validated algorithm that examines the gyroscope data from the AX6, comparing the output metrics to the output from the previously used Opal sensor.

The bland-Altman plots need to be made clearer to reflect agreement or not along with the quantitative measures.

The Bland-Altman plots provide a visual overview of the quantitative data that is reported in Table 2. The limits of agreement and mean difference are reported in the table and are visible in the plots.

Table 2, the Pearson correlations, make headings consistent.

This has been amended.

The authors need proper justification for the comparison with AX6 device. Simply stating the cost is not reflective of the quality of the measurement produced by the device. Trying more devices would add more reliability to the evaluation.

The AX6 wearable sensor (~£186) is substantially lower cost that the Opal device (~£2000 per sensor). Device cost is a huge factor in the adoption of technologies within clinical trials and healthcare settings. While there are wearables that may be lower cost or that may provide similar quality data, the Axivity sensors (AX3 and AX6) have been extensively used for mobility assessments within various clinical groups, being CE marked and applicable for clinical trials / research. Therefore, we have focused on this sensor to validate the turning algorithm, as we can add turning output to our previous gait and mobility validation work with the same sensor (Powell et al. 2021), which will mean multiple validated outputs from the same sensor in the same location.

Round 2

Reviewer 2 Report

Authors clearly answered my comments. I have noo further comments.

Author Response

We thank the reviewer for their kind comment.